# Oncogenic Viruses-Encoded microRNAs and Their Role in the Progression of Cancer: Emerging Targets for Antiviral and Anticancer Therapies

**DOI:** 10.3390/ph16040485

**Published:** 2023-03-23

**Authors:** Mahmoud Kandeel

**Affiliations:** 1Department of Biomedical Sciences, College of Veterinary Medicine, King Faisal University, Al-Ahsa 31982, Saudi Arabia; mkandeel@kfu.edu.sa; 2Department of Pharmacology, Faculty of Veterinary Medicine, Kafrelsheikh University, Kafrelsheikh 33516, Egypt

**Keywords:** oncogenic viruses, viral miRNAs, cancer, human neoplasms

## Abstract

Approximately 20% of all cases of human cancer are caused by viral infections. Although a great number of viruses are capable of causing a wide range of tumors in animals, only seven of these viruses have been linked to human malignancies and are presently classified as oncogenic viruses. These include the Epstein–Barr virus (EBV), human papillomavirus (HPV), hepatitis B virus (HBV), hepatitis C virus (HCV), Merkel cell polyomavirus (MCPyV), human herpesvirus 8 (HHV8), and human T-cell lymphotropic virus type 1 (HTLV-1). Some other viruses, such as the human immunodeficiency virus (HIV), are associated with highly oncogenic activities. It is possible that virally encoded microRNAs (miRNAs), which are ideal non-immunogenic tools for viruses, play a significant role in carcinogenic processes. Both virus-derived microRNAs (v-miRNAs) and host-derived microRNAs (host miRNAs) can influence the expression of various host-derived and virus-derived genes. The current literature review begins with an explanation of how viral infections might exert their oncogenic properties in human neoplasms, and then goes on to discuss the impact of diverse viral infections on the advancement of several types of malignancies via the expression of v-miRNAs. Finally, the role of new anti-oncoviral therapies that could target these neoplasms is discussed.

## 1. Introduction

Early in the twentieth century, researchers made their first discoveries suggesting that cancer may have an infectious cause such as a viral infection (Figure 1). Ellermann and Bang in 1908 and Rous in 1911 discovered that cell-free tumor extracts might transmit avian leukemias and sarcomas in birds, which pointed to a viral etiology [1]. The Epstein–Barr virus (EBV) was discovered by Sir Anthony Epstein, Bert Achong, and Yvonne Barr in cell cultures taken from a child with Burkitt lymphoma [2].

Approximately 20% of all cases of human cancer are caused by viruses [3]. Although a great number of viruses are capable of causing a wide range of tumors in animals, only seven have been linked to human malignancies and are presently classified as oncogenic viruses. These include the human papillomavirus (HPV), Epstein–Barr virus (EBV), hepatitis B virus (HBV), hepatitis C virus (HCV), human herpesvirus 8 (HHV8), human T-cell lymphotropic virus type 1 (HTLV-1), and Merkel cell polyomavirus (MCPyV). Hepatocellular carcinoma (HCC) is the most prevalent kind of liver cancer, and HBV and HCV are the causes of approximately 90% of all cases of HCC [4]. Besides being the principal cause of cervical cancer and other anogenital neoplasms, high-risk HPV strains are also responsible for a significant percentage of head and neck cancers. Hodgkin lymphoma, Burkitt lymphoma, and nasopharyngeal carcinoma have all been associated with EBV. Kaposi sarcoma, caused by the Kaposi sarcoma-associated herpes virus (KSHV), is a cancer found most often in persons with acquired immunodeficiency syndrome (AIDS). The Merkel cell cancer that is caused by MCPyV and the adult T-cell lymphoma that is caused by HTLV-1 may be found in hosts [5].

Viruses employ a variety of tactics to achieve their ultimate objectives, which include survival, proliferation, and transmissibility. It is possible that virally encoded microRNAs (miRNAs), which are ideal non-immunogenic tools for viruses, play a significant role in these processes. miRNAs are able to modulate the gene expression of both the virus and the host target, and this can ultimately result in the immune invisibility of infected cells [6]. It appears that viral miRNAs (v-miRNAs) play a significant role in the survival and proliferation of viruses by implementing various mechanisms of immune evasion. Both v-miRNAs and host-derived miRNAs (host miRNAs) have the ability to influence the expression of various host- and virus-encoded genes (Table 1, Figure 2) [7]. In DNA viruses, v-miRNAs, such as cellular miRNAs, are transcribed in the nucleus as primary v-miRNAs (pri-v-miRNAs). The DROSHA enzyme then cleaves pri-v-miRNA into pre-v-miRNAs. The pre-v-miRNA is exported to the cytoplasm by exportin 5. Cleavage by the DICER enzyme produces mature v-miRNA from pre-v-miRNA. miRNA is imported into the RISC complex to target the viral or cellular mRNA. The RISC complex then cleaves the targeted mRNA. Because most RNA viruses reproduce in the cytoplasm, viral RNAs are unable to interface with nuclear miRNA processors; this may explain why there is so little evidence that RNA viruses produce functional miRNAs [8]. However, DROSHA was shown to translocate into the cytoplasm after infection with viruses, allowing for the perfect synthesis of miRNAs [9].

An intriguing concept suggests that it may be promising to use the cellular miRNAs of v-miRNA orthologues; they control the same targets as they have the same seed sequence. This hypothesis was supported by several lines of evidence. In this regard, there was a high homology rate between the sequences of several viral and host miRNAs, which implied that they had similar downstream activity [10]. The present literature review starts with the explanation of how viral infections exert their oncogenic characteristics in human neoplasms and continues with a discussion of the effect of various viral infections on the progression of several kinds of malignancies through the expression of v-miRNAs.

## 2. Oncogenic Viral Infections

There are different types of DNA and RNA viruses that are categorized as oncogenic. They cause cancer by a number of methods, one of which is the encoding of miRNAs (Figure 3). Several of the viruses known to be involved in cancer induction or progression are described below.

The herpesviruses family includes EBV. This virus is an enveloped virus that, similar to other herpesviruses, has a DNA core surrounded by an icosahedral nucleocapsid and an associated tegument. It is now common knowledge that over 90% of the adult population around the world have contracted EBV at some point, and once they have been infected, they continue to harbor the virus throughout their lives [26]. An exchange of saliva is the mode of transmission for EBV. This oncogenic virus does most of its infecting and replicating in the stratified squamous epithelium of the oropharynx during an acute infection [27,28]. After this, the B cells become infected with a dormant form of the virus (although the sequence of epithelial versus lymphoid infection is a matter of debate). It is believed that the infection of B lymphocytes by EBV takes place in the lymphoid organs of the oropharynx, and in normal carriers, the virus is assumed to survive in circulating memory B cells [29]. Research on epidemiology and immunology suggests that EBV infection may have a role in the development of endemic Burkitt lymphoma. The epithelial cancer known as nasopharyngeal carcinoma is connected with EBV and is prevalent in South China and Southeast Asia [30]. More than 97% of patients with nasopharyngeal carcinoma have a positive EBV test, and a close connection between nasopharyngeal carcinoma and EBV is observed all over the world [31]. In addition to infection, EBV can cause a prevalent kind of cancer called EBV-associated gastric cancer [32], which has a regional distribution of 1.3% to 30.9% of all cases of gastric cancer and a global average of 8.9% of all cases of gastric cancer, or approximately 75,000 new cases each year [33]. Because EBV is associated with the production of certain subsets of latent proteins, the establishment of a latent infection by EBV has been linked to a number of different types of cancer. Epstein–Barr nuclear antigens (EBNAs) 1, 2, 3A, 3B, 3C, and LP are expressed during latent infection in host cells [34]. EBNAs are multi-functional proteins and considered as the only proteins expressed in all types of EBV and found in EBV-associated malignancies [35]. The positive and negative regulation of viral promoters by EBNA1 is essential for gene control, extrachromosomal replication, and maintenance of the EBV episomal genome [36].

The human cytomegalovirus (HCMV) is a member of the Herpesviridae family and has a DNA genome that is 236 kilobase pairs in size [37]. HCMV gene products interfere with cell cycle progression, promote mutation and genomic instability in the viral genome, enhance cell survival, and facilitate immune evasion and tumor growth [38]. In addition, HCMV can infect a wide variety of cell types found in tumors and their microenvironments. HCMV-induced oncomodulation was predicted after the discovery of viral proteins and DNA in several cancer tissues [39]. HepG2 cells that had been infected with HCMV, for instance, were observed to secrete IL-6 in conjunction with autocrine and/or paracrine activation of the IL–6R–JAK–STAT3 pathway. In HepG2 cells that had been infected with HCMV, an increase in cell proliferation concurrently occurred with an increase in the synthesis of cyclin D1 and survivin. Compared with cultures that were not infected with HCMV, cultures that were infected with HCMV had an increase in the development of tumorspheres in HepG2 cells [40]. In addition, when the herpes simplex virus type 1 (HCMV) infected the “stem-like” colorectal cancer HT29 and SW480 cells, both the epithelial–mesenchymal transmission (EMT) pathways and the WNT pathways were activated, which led to increased cellular proliferation and motility [41].

The Papillomaviridae family includes HPV. The DNA genome is double-stranded and contained in non-enveloped virions that make up the virus. An icosahedral capsid that is comprised of the major and minor structural proteins L1 and L2, respectively, encloses the genetic material. These viruses have a very precise tissue preference and can infect the epithelium of the skin as well as the mucosa. On the basis of the genomic sequence of the L1 gene, which codes for the primary capsid protein, over 200 different forms of HPV have been discovered and characterized [42]. Of these, at least 14 high-risk types have the potential to cause cancer. Two kinds of HPVs are responsible for the majority of HPV-related malignancies (productive for CIN1 or abortive for CIN3), including about 70% of cervical cancers and precancerous cervical lesions [43]. Integration often ends in the deregulation of the expression of the viral E6 and E7 oncogenes, which in turn stimulates cellular proliferation, eliminates cell cycle checkpoints, and ultimately leads to increasing genetic instability. This provides cells with an advantage in selective proliferation and increases the propagation of oncogenic mutations [44]. The significance of abnormally dysregulated oncogene expression is shown by the clonal expansion of cells that have incorporated HPV. In fact, the continuing growth and survival of HPV-associated malignancies is contingent on the viral E6 and E7 oncogenes being expressed [45].

HBV, which belongs to the family Hepadnaviridae, has a genome that is 3.2 kilobase pairs in size and contains partly double-stranded DNA. There are four key open reading frames (ORFs) in the HBV genome. These ORFs encode for the polymerase; the surface protein HBsAg, a core protein that forms the nucleocapsid; and the HBV X protein, which is critical in viral replication [46,47]. In eastern Asia and sub-Saharan Africa, infection with endemic HBV is the primary cause of HCC, accounting for around 70% of all cases.

The prevalence of HCV infection in the nations of Europe and North America varies from 50% to 70%, whereas alcohol abuse, which can lead to alcoholic steatohepatitis (ASH), is responsible for around 20% of all cases [48]. The development of HCC typically occurs as a result of a protracted and chronic disease course that is accompanied by underlying liver cirrhosis (~80%) [49]. However, approximately 20% of HCC cases develop in livers that do not have cirrhosis. Only sexual contact with blood or other bodily fluids, or vertical transmission from mother to child, is capable of transmitting HBV, and only a small number of HBV virions are needed to initiate an infection. Most chronic carriers acquire their status because of infection during pregnancy, delivery, or early childhood, when the immune system is still developing. Between 1% and 5% of adults and adolescents who contract an infection will develop a chronic condition [50].

The polyprotein encoded by the HCV is translated and then cleaved into structural (S) and non-structural (NS) proteins. The HCV is a single-stranded, positive-sense RNA virus. Some examples of structural proteins are the core protein, the envelope E1 and E2 glycoproteins, and the p7 protein. Non-structural proteins (NS1, NS2, NS3, NS4A/B, and NS5A/B) aid in viral genome replication and particle assembly. Because it does not encode oncoproteins and integrate its genome into the chromosomal DNA of the host, HCV is unique among cancer-causing viruses. The hepatocyte cytoplasm is the replication site for HCV [51]. It was previously believed that HCV-related HCC development occurred primarily through indirect mechanisms, such as the effects of chronic inflammation and oxidative stress, because HCV RNA cannot integrate into the human genome. Eventually, this condition leads to fibrosis and cirrhosis, as is the case with other causes of HCC such as ASH, non-alcoholic steatohepatitis (NASH), and obesity-related illnesses. The viral proteins themselves, however, have been demonstrated to have a direct oncogenic impact, according to recent studies [52].

The hepatitis delta virus (HDV) is a small circular single-stranded RNA satellite virus that is dependent on HBV for progeny virus production. The HBV envelope glycoprotein is required for HDV particle assembly. As a result, HDV can only produce an infection if there is also an HBV infection present. The most severe form of viral hepatitis is caused by HDV, which affects between 15 and 20 million people throughout the world [53].

It has been proven that immunodeficiency is a risk factor for the development of cancer, and it is probable that the underlying causes are numerous. Some of these possible explanations include the uncontrolled growth of oncogenic viruses and insufficient immune surveillance. The presence of CD4 T cells, as well as their quantity and functionality, are critical components in multiple stages of the oncogenic pathway [54]. These stages include the recognition of tumor antigens, the production of an antibody capable of effectively neutralizing the tumor, cellular responses to viral pathogens, and the elimination of premalignant lesions. Human immunodeficiency virus (HIV-1) carriers have an increased risk of developing a number of cancers, including non-Hodgkin lymphoma, Kaposi sarcoma, cervical cancer, and other cancers connected to chronic viral infections. This has historically been linked to immunological suppression brought on by HIV-1, which is characterized by a decline in CD4+ T-helper cells, exhaustion of lymphopoiesis, and defective lymphocytes. Antiretroviral treatment initiated early could not prevent the development of oncologic problems in the long term. This showed that HIV-1 and its antigens are actively involved in carcinogenesis and may have an influence even at extremely low levels on the background of a rebuilt immune system [55].

The Kaposi sarcoma tissue that was initially examined for the presence of KSHV or HHV-8 was obtained from individuals who suffered from AIDS. There is a correlation between having an infection with KSHV and having certain inflammatory diseases. KSHV has been found to be present, in addition to KS, in primary effusion lymphoma, KSHV-associated lymphoma, and some instances of multicentric Castleman disease [56]. The majority of original KSHV infections do not produce any clinical signs, and similar to other cancers resulting from human oncogenic viral infections, Kaposi sarcoma does not manifest itself until decades after the virus has been dormant. In addition, asymptomatic oral shedding and transmission through body fluids are both possible modes of KSHV transmission [57].

MCV is thought to be the cause of Merkel cell carcinoma, which is an uncommon but aggressive form of skin cancer [58]. The sequencing of the virus recovered from Merkel cell tumors contains tumor-specific mutations that terminate the MCV T antigen. These mutations (which are not present in the wild-type virus recovered from non-tumor sites) inactivate the T antigen helicase, leaving the integrated virus incapable of replicating outside of the host cancer cell. As a result, the tumor acts as a “dead-end host” for MCV [59].

HTLV-1 is a retrovirus associated with several diseases, including lymphomas and myelopathies. It is thought that 1% to 5% of HTLV-1-infected cases develop cancers [60], such as. adult T-cell lymphoma/leukemia (ATLL) and cutaneous T-cell lymphoma.

## 3. Viral miRNAs and Human Cancers

### 3.1. EBV

A number of human cancers, including the endemic Burkitt lymphoma and nasopharyngeal carcinoma, have been linked to EBV, which is a gamma-herpesvirus [61]. It was reported in 2004 that EBV was one of the first known human oncoviruses to encode v-miRNAs. It was also found that these molecules interacted with viral and host targets. EBV miRNAs have an effect on critical biological processes involved in carcinogenesis, and as a result, they contribute to the transformation of cells and the growth of tumors in EBV-associated cancers [61].

In a study by Hsu and colleagues, the miRNA known as miR-BART9 was found to be associated with the enhancement of cell motility and invasiveness in cultured nasopharyngeal cancer cells. It was shown that the promigratory activity observed in vitro translated into an improved capacity for metastatic disease spread in vivo. Computational investigation suggested that miR-BART9 became bound to and inhibited the function of the membrane protein E-cadherin. E-cadherin is crucial for the maintenance of cell–cell junctions and the epithelial phenotype. They were able to demonstrate that miR-BART9 specifically suppressed E-cadherin, which in turn caused nasopharyngeal cancer cells to exhibit a phenotype characteristic of mesenchymal cells and promoted cell migration. The results of this study confirmed that miR-BART9 is a prometastatic v-miRNA and also provided evidence that miR-BART9 overexpression in EBV-positive nasopharyngeal cancer cells may contribute to tumor cell aggressiveness [11]. Furthermore, it has been shown that EBV-miR-BART1 was strongly expressed in nasopharyngeal cancer and was directly related to patients’ advanced stages. By specifically targeting PTEN, BART1 accelerated nasopharyngeal cancer cell migration and invasion in vitro and accelerated tumor metastasis in vivo. A reduction in PTEN dosage by BART1 activated PTEN-dependent pathways, induced EMT, and boosted nasopharyngeal cancer migration, invasion, and metastasis [12]. BART10-3p may encourage tumorigenicity in vivo and in vitro dedifferentiation, EMT, and proliferation of nasopharyngeal cancer cells [13]. The 3′UTR of ALK7 was directly targeted by BART10-3p, which inhibited its expression. Restoration of ALK7 reversed the malignant phenotypes that BART10-3p caused [13].

Patients with stomach cancer linked to EBV had poor survival rates when miR-BART20-5p was highly expressed. The fact that EBV BART miRNAs were easily found in EBV-associated gastric cancer tumor tissues, as opposed to the paired normal tissues, seems notable [14]. Additionally, miR-BART2-5p and miR-BART11-5p inhibited apoptosis and enhanced S-phase arrest of the cell cycle, while increasing the migration and proliferation of gastric cancer cells. Therefore, via targeting RB and p21, BART2-5p and BART11-5p played significant roles in increasing proliferation and migration and preventing apoptosis in EBV-associated gastric cancer [15]. It is established that miR-BART5-5p controls the miR-BART5/PIAS3/pSTAT3/PD-L1 axis and directly targets PIAS3 while enhancing PD-L1 in PD-L1(+) tumors. Overall survival in EBV-associated gastric carcinoma patients was significantly shortened for those with PD-L1(+) tumors compared with those with PD-L1(−) tumors. This indicated the general efficiency of miR-BART5-5p in PD-L1(+) tumors and its low efficiency in PD-L1(−) tumors, implying the importance of PD-L1(+) in the overall actions of miR-BART5-5p. Based on this finding, miR-BART5-5p could be a PD-1/PD-L1 immune checkpoint inhibitor therapy [16]. The immune checkpoint inhibition promotes antiapoptosis and tumor cell growth, invasion, and migration, as well as immune evasion, which accelerates the course of gastric cancer and worsens the clinical result [16]. In addition, the EBV-encoded viral miRNA-BART11 downregulated the FOXP1 transcription factor and promoted EMT through acting on gastric tumor cells and the tumor microenvironment. This could hasten the invasion and spread of cancer, which would have repercussions for patient survival and prognosis [17]. Reduced levels of FOXP1 led to changes in the expression of the EMT transcription factors E-cadherin and snail. The EBV-miR-BART11-FOXP1 signaling axis could enhance the activity of EMT inducers NF-κB and IL-6 through the downregulation of E-cadherin. Additionally, it has been observed that EBV-miR-BART18-3p changed the lipogenesis pathway during EB virus injection, which promoted and contributed to colorectal cancer metastasis [18].

There was an increase in viral miRNAs in Burkitt lymphoma patients. The v-miRNAs BART1–3p, BART3, BART4, BART6–3p, BART6–5p, BART7, BART8, BART9, and BART10 and 12 other EBV miRNAs were upregulated in Burkitt lymphoma. These results bolstered the theory that the expression pattern of EBV-related miRNAs in Burkitt lymphoma is disease-specific and offered insight into the possible function of EBV-encoded miRNA in this cancer [62]. miR-BART3 was found to target the importin 7 receptor, which transports transaction factors to the nucleus [63]. The miR-BART6-3p can target the host immune response by interfering with type I IFN pathways [64]. The miR-BART2-5p prevented the virus from being attacked by immune cells by inhibiting the MHC-class 1 chain-related molecule B [65].

Several miRNAs were found to be associated with the progression of Hodgkin lymphoma. miR-BART2-5p was upregulated in Hodgkin lymphoma; it inhibited NF-kB activation and helped in the immune evasion of cancer cells [66]. The exosomal release of miR-BART13-3p, the most common miR-BART in Hodgkin lymphoma, has been shown to have a role in the development of the disease [67].

### 3.2. HCMV

Several miRNAs encoded by HCMV affect the expression of viral and cellular genes that are involved in viral replication, the generation of cytokines, cell survival, and both the innate and adaptive immune responses. Because of their ability to silence human genes involved in a variety of physiological processes, HCMV-miRNAs have lately emerged as being prospective clinical causes of a wide range of human disorders [68]. Initial reports showed that HCMV was not a virus that causes cancer, so it seems likely that HCMV will profit from the degradation of host miRNAs for reasons unrelated to oncogenesis [69]. However, oncogenic characteristics are possessed by multiple members of the HCMV proteome [70]. It has been demonstrated that the expression of the HCMV chemokine receptor US28 in NIH3T3 cells resulted in transformed phenotypes and tumor formation [71]. The HCMV UL76 protein has also been associated with chromosomal breaks in human glioblastoma cells [72].

Reduced cytotoxic T-cell immune responses were caused by the targeting of HCMV miR-US4-selective 1 of ERAP1, which in turn stifled the development of precursors into mature MHC class I-presented epitopes [19]. CMV70-3P miRNA has been demonstrated to boost the stemness of glioblastoma multi-form cancer stem cells. The ability of primary glioma cells to proliferate and produce neurospheres was drastically reduced by blocking CMV70-3P expression with oligo inhibitors. CMV70-3P has been shown to increase cellular SOX2 expression at the molecular level [20]. RhoA, a small GTPase necessary for CD34+ HPC self-renewal, proliferation, and hematopoiesis, is confirmed to be a target of HCMV miR-US25-1. MiR-US25-1 expression reduces signaling via the non-muscle myosin II light chain, which stops cytokinesis and inhibits proliferation. Additionally, infection with an HCMV mutant missing miR-US25-1 caused CD34+ HPCs to proliferate more and the proportion of cells with genomes at the end of a latency culture to drop [73].

The antiapoptotic effects of HCMV-miR-UL70-3p in HEK293T cells have been studied by Pandeya et al. [74]. They showed that HCMV-miR-UL70-3p interacted with the 3′UTR of the mRNA for the proapoptotic gene modulator of apoptosis-1 (MOAP1). Ectopic expression of an HCMV-miR-UL70-3p mimicked greatly attenuated H2O2-induced apoptosis by inhibiting MOAP1 translation. The antiapoptotic effect of HCMV-miR-UL70-3p was bolstered by suppressing MOAP1 with siRNA. Reductions in apoptosis caused by H2O2 were also observed [75]. The essential EGFR adaptor protein GAB1, which triggers cellular proliferation in response to EGF and activates and maintains signaling via the PI3K and MEK/ERK pathways, has also been shown to be directly suppressed by HCMV miR-US5-2 [74].

### 3.3. KSHV

The human pathogenic γ-herpesvirus known as KSHV is closely linked to the occurrence of primary effusion lymphoma, as well as KS and B-cell proliferative diseases. A topic of rapidly growing importance is the documentation and mechanistic analysis of non-coding RNAs expressed by KSHV. miRNAs are expressed by KSHV during both the latent and lytic phases of infection [76,77]. The control of these miRNAs may change at various points in the course of an infection. KSHV expresses a subset of miRNAs during the latent phase of infection that aid the virus in evading the host immune response and staying dormant. The viral latency-associated nuclear antigen (LANA) protein regulates these miRNAs by binding to the promoter regions of the miRNA genes [78]. A distinct set of miRNAs implicated in viral replication and pathogenicity is expressed by KSHV during the lytic phase of infection [77]. The viral immediate early gene product lytic replication and transcription activator (RTA) controls the expression of miRNAs by binding to promoter sites and turning them on. Therefore, pathways that either promote or inhibit lytic replication are crucial to the pathogenicity of KSHV. The RTA, together with the LANA, are thought to control this delicate equilibrium [79].

Particular attention has been paid to KSHV miRNAs generated from 12 stem-loops found in the main latency locus [80]. It was discovered that all KSHV miRNAs induced a greater release of the VEGFA protein into the endothelial cell supernatant and upregulated the production of the MMP1, MMP13, VEGFA, and VEGFR2 transcripts. Based on these results, it is clear that KSHV miRNAs are involved in controlling the levels of angiogenic factors and matrix metalloproteinases (MMPs) [81].

Patients with HIV-related KS have a worse prognosis if they have high levels of KSHV-miR-K12-1, which is increased in gastrointestinal KS tissues. The transfection of a miR-K12-1 inhibitor led to a significant drop in BCBL-1 cell viability and an increase in cell death compared with the control group, while transfection of a miR-K12-1 mimic led to an increase in cell proliferation and mitosis. In addition, miR-K12-1 stimulated cell proliferation in HIV-related gastrointestinal KS by activating the PI3K/Akt pathway, and LY294002 inhibited this effect [82]. Furthermore, several KSHV miRNAs targeted GADD45B for suppression. They demonstrated that Nutlin-3, a p53 activator, could trigger apoptosis via repressing GADD45B through KSHV miRNAs. Both the ectopic expression of GADD45B and an antisense inhibitor of a particular KSHV miRNA increased apoptosis in the context of KSHV infection. These findings suggested that KSHV miRNA has a variety of roles, some of which involve controlling DNA damage response proteins to support the survival of infected cells in the presence of stress signals [83]. KSHV miR-K6-5p and the tumor-suppressing cellular miR-15/16 miRNA family share similar sequences. A hallmark activity of miR-16, the suppression of cell cycle progression by miR-K6-5p, is shown to exist. Targets of miR-K6-5p included several cell cycle regulators that are conserved among members of the miR-15/16 family [84].

A number of cancers, such as breast, prostate, lung, and kidney cancers, have been found to have abnormal signaling of the mammalian target of rapamycin complex 1 (mTORC1) [85]. mTORC1 is inhibited by the cytosolic arginine sensor for mTORC1 (CASTOR1 and CASTOR2). The oncogenic mTORC1 pathway is typically deregulated in KS. This pathway controls cell proliferation, survival, and metabolism. Research has demonstrated that cancer cells may alter CASTOR1 and CASTOR2 to promote carcinogenesis through regulating mTORC1 signaling. The miRNA miR-K4-5p, and perhaps miR-K1-5p, expressed by KSHV, specifically targeted and repressed CASTOR1 expression. When mTORC1 activation was suppressed by knocking down miR-K1-5p and miR-K4-5p, CASTOR1 expression was re-established. Both mTOR inhibitors and the overexpression of CASTOR1 or CASTOR2 were able to counteract the effects of pre-miR-K1 and pre-miR-K4 on mTORC1 activation and the growth transformation they induced. Based on our findings, it seems reasonable to target the mTORC1 pathway in the treatment of KSHV-associated malignancies [86]. Additionally, Li et al. demonstrated that miR-K3 and the signal pathway it creates played a role in KSHV latency and angiogenesis generated by KSHV. They discovered that miR-K3 overexpression caused angiogenesis as well as viral latency by blocking viral lytic replication [21].

KSHV-miR-K12-1-5p can prevent cell apoptosis while promoting the proliferation, migration, and invasion of KS cells. KSHV-miR-K12-1-5p has been found to directly target the suppressor of cytokine signaling 6 (SOCS6), and it has the ability to inhibit SOCS6 expression. Additionally, the impacts of the KSHV-miR-K12-1-5p suppressors were reversed by SOCS6 knockdown [22]. It has been shown that miR-K12-1 acted as an oncogene by hijacking IκBα/NF-κB signaling to cause STAT3 to be activated in response to IL-6. Additionally, it was discovered that the new miR-K12-11/I-B/NF-B/IL-6/STAT3 oncogenic signaling pathway significantly participated in the development of KSHV tumors [87].

### 3.4. HPVs

One of the most prevalent sexually transmitted illnesses, HPV infection, is linked to malignancies such as cervical, anal, and head and neck squamous cell carcinomas. More than 200 different kinds of HPV have been found so far [88,89]. According to their propensity to cause cancer, various mucosal HPV types are categorized as either high-risk HPV/oncogenic HPV types, such as HPV16, 18, 31, and 33, or low-risk HPV/non-oncogenic HPV types, such as HPV6 and 11, which are commonly observed in warts [90].

It has been demonstrated that several virions, including various HPV genotypes, have the ability to code miRNA-like species. These small molecules may play a role in the development of virally induced cancer. However, the data available on the role of HPV-encoded miRNAs in cancer are scarce; one reason for this is likely to be the lack of appropriate study models for the many HPV strains [91]. Chirayil et al. have used a novel method for miRNA detection based on forced genome expression to describe novel HPV-encoded miRNAs. It is worth noting that FcPV1-miRNAs play a role in managing the HPV virus’s life cycle [92]. Further investigations by Virtanem et al. reported that tumor samples included miRNAs from the HPV-16 species miR-H1, miR-H3, miR-H5, and miR-H6 [93]. Qian et al. sequenced human cervical lesion tissue and cell lines, and discovered nine miRNAs that were likely to be encoded by HPV (HPV6-mir-H1, HPV16-mir-H1, HPV16-mir-H2, HPV16-mir-H3, HPV16-mir-H5, HPV16-mir-H6, HPV38-mir-H1, HPV45-mir-H1, and HPV68-mir-H1). These miRNAs were upregulated in chronic infections, and their interference with many pathways, such as cell cycle regulation, the immune response, and cell adhesion and migration, may have contributed to the tissue’s propensity to change [23]. It is interesting to note that the HPV 16 transcriptional enhancer factor (TEF-1) controlled cell migration and proliferation and bound to and triggered the E6 and E7 promoters of early HPV 16 [94]. In this context, TEF-1 and its associated factors were essential for HPV 16 transcription.

### 3.5. Hepatitis Viruses (HBV and HCV)

The HBV is a small, non-cytopathic, hepatotropic DNA virus. The presence of HBV infection is one of the most important risk factors for chronic hepatitis, cirrhosis of the liver, and hepatocellular carcinoma across the world [95]. Hepatocellular carcinoma tissue was shown to contain small ncRNAs that were transcribed from the HBV genome [96]. HBV-miR-3, an HBV-encoded miRNA, influenced HBV replication. By preventing PPM1A translation, HBV-miR-3 rendered it inaudible. PPM1A downregulation boosted cell proliferation, which was linked to the development of hepatocellular carcinoma [24]. HBV-encoded miRNAs are attractive therapeutic targets because they are virus-specific and can be used to control HBV transcripts and, thus, viral output [97]. These HBV-encoded miRNAs may also regulate cellular proteins that promote HBV replication, immune evasion, signaling, or excretion. Additionally, researchers detected three putative hairpin-like structures in liver biopsy tissue by small RNA NGS, which they termed HBV-miR-6, HBV-miR-7, and HBV-miR-8. Only HBV-miR-6 could interact with RISC among these putative HBV-encoded miRNAs [97].

The HCV was originally recognized as a non-A, non-B hepatitis. It is a single-stranded, positive-sense RNA virus from the genus Hepacivirus and family Flaviviridae [98]. Worldwide, HCV infection is still considered to be a significant risk factor for chronic hepatitis, liver cirrhosis, and hepatocellular cancer. The development of HCV infection and its spread in infected hepatocytes and role in the development of liver fibrosis depend heavily on miRNAs [99]. They attack the vital host cellular components required for successful HCV replication and accelerated cell growth [100].

A study by Yang et al. demonstrated that HBV-encoded miRNA3 (HBV-miR-3) was expressed in HBV-infected tissues. To control HBV replication, HBV-miR-3 was transcribed during infection. This miR-3 regulated HBV replication by inhibiting HBsAg and HBeAg transcript synthesis [101], as well as targeting the DIX domain containing 1 (DIXDC1) and protein phosphatase 1A (PPM1A) [24].

It is believed that HCV does not encode v-miRNAs [102]. In HCV, the host miRNAs play a central role in virus replication and constitute a valid antiviral target. Initially, the role for miR-122 in HCV infection was indicated by sequestering endogenous miR-122, which led to a significant decrease in HCV RNA in human liver cells that contained HCV replicons [103]. This was due to the interaction of miR-122 with two neighboring binding sites, both of which had seed matches that were complementary to the miRNA [104]. It was interesting to find that these binding sites were located in the viral RNA’s 5′ UTR. They were shared by all HCV genotypes and found in a single-stranded region of RNA just upstream of the HCV IRES. Both miR-122 binding sites must be present for effective HCV replication. Site 1 is completely necessary for infection in an infectious HCV system, whereas overexpression of miR-122 can eliminate the need for site 2 [104,105]. Similar to miR-122, mir-199a* has been found to bind to the HCV genomic RNA. It has been established that miR-199a* inhibits HCV RNA replication by binding to a region of the HCV 5′ UTR that is conserved across all HCV genotypes [106]. miR-199a* serves as a viable antiviral target against HCV. The overexpression of miR-199a* has antiviral action by inhibition of viral genome replication [99].

### 3.6. HTLV-1 and MCPyV

HTLV-1 oncogenic mechanisms directly impacted viral proteins in the host miRNA machinery. In this regard, HTLV-1-encoded miRNAs have not been reported [107,108]. HTLV-1 increased the expression of host cell miRNAs involved in proliferation and apoptosis [108]. Comparison of the miRNA expression profiles in normal and HTLV-1-infected hosts revealed several deregulated miRNAs, including miR-29c, miR-30c, miR-193a-5p, and miR-885-5p, which could be regarded as diagnostic tools for HTLV-1 infection [109]. It was reported that approximately 22 upregulated and 22 downregulated miRNAs were identified in acute ATLL patients [107].

Reports have indicated that MCPyV miRNAs were either not present or were present at extremely low levels in malignancies [110]. MCV-miR-M1-5p, which is MCPyV-encoded miRNA, was found to be expressed at low levels in about half of the virus-positive cases. Interestingly, MCV-miR-M1 has been associated with negative regulation of viral DNA expression [111].

## 4. miRNA-Based Therapy for Oncogenic Viruses

The use of miRNAs and anti-miRNA oligonucleotides (AMOs) as potential new treatments has been proposed [112]. miRNAs account for only approximately 3% of human genes, yet they may influence up to 30% of human genes that code for proteins. One intriguing therapeutic idea is that a single miRNA can negatively affect several target proteins by interacting with various target mRNAs [113]. Normalizing the aberrant levels of miRNAs could result in the recovery of affected cells, prevent tumor differentiation, and prohibit cancer metastasis [113]. In this context, miRNA expression alterations could be dealt with in the following two primary ways: (1) degradation of the overexpressed mRNA via miRNA, and (2) regulation of miRNA expression. Injecting miRNAs to make up for the inadequate production of miRNAs is a feasible option.

Yang et al. (2013) devised a plan for combatting HCV infection that involves inserting sequences from the HCV genome into five endogenous miRNAs from a naturally occurring miRNA cluster (miR-17-92). This miRNA cluster (HCV-miR-Cluster 5) was supplied to cells using adeno-associated virus (AAV) vectors, and it made sense that it would be assembled in the liver, where HCV replication and assembly take place. AAV-HCV-miR-Cluster 5 was able to reduce HCV multiplication by 95% in 2 days in vitro, and the steady production of anti-HCV miRNAs blocked HCV from spreading to uninfected cells [114].

### 4.1. Anti-miR

With respect to tumor progression, miRNA can be viewed as having two classes, tumor-suppressor miRNA and tumor-inducer miRNA. The overexpression of tumor-enhancing miRNA can be overcome by the application of anti-miR. This can be accomplished by the administration of the artificial antisense oligonucleotide. One example of an upregulated miRNA in an oncogenic viral infection is miR149 in HCV infection [115]. The first anti-miR compound introduced to clinical trials was miravirsen, which was used to treat HCV by targeting the liver miR122 [116]. MiRNA sponges are a novel class of miRNA inhibitors, comprising transcripts with tandem repeats of specific sequences for binding a desired miRNA [117]. They act by sequestering the overexpressed miRNA.

EBV BART miRNAs are important targets for treating EBV-associated epithelial tumors because they are overexpressed in these cases [14]. Gold nanoparticles bearing anti-EBV-miR-BART7-3p were successful in decreasing the growth of cancer cells [118].

Anti-miRs are useful because they can be directed against specific miRNAs that play a role in controlling EBV gene expression, replication, and latency. There are several examples of viral miRNAs that have been proven to promote EBV replication and carcinogenesis, including EBV miR-BART6-3p and EBV miR-BART16-5p, miR-BHRF1-2-5p, miR-BART1, BART2, and BART22. As a potential treatment for EBV-associated malignancies, anti-miRs targeting these miRNAs can be created and evaluated in preclinical trials.

Some studies have reported the upregulation of certain miRNAs in HPV-associated cancers. For example, in cervical cancer associated with high-risk HPV types such as HPV-16 and HPV-18, miR-9, miR-21, miR-27b and miR-34a [119] have been reported to be upregulated.

Studies have shown that some of the HHV-8-encoded miRNAs were upregulated in cancer cells, suggesting that they may have contributed to the development and progression of these cancers. For example, the HHV-8 miRNA known as miR-K12-11 has been shown to promote the growth and survival of cancer cells by targeting tumor suppressor genes [120]. A specific anti-miR specific for miR-K12-11 can be a promising target against oncogenic herpesviruses.

### 4.2. miR Mimetics

miR mimetics are synthetic molecules designed to mimic the function of endogenous miRNAs. These molecules can be used to regulate the expression of specific genes and have potential applications in the treatment of various diseases. The lowered expression of certain miRNAs can enhance the tumorigenic environment. In such therapies, miR mimetics are usually developed for the downregulated host miRNAs. In this case, treatment occurs by the compensation of lowered levels of these miRNAs [117]. Examples of downregulated miRNAs in oncogenic HCV infections include miR29a, miR29b, miR29c, miR17, miR106a, miR106b, miR181a, miR93, miR221, and miR222 [121]. In another example, HCV replication was adversely affected by the overexpression of miR-199a in its binding to the viral 5′UTR [106].

miR-28-3p was shown to target a site inside the HTLV-1 viral gag/pol mRNA, where it suppressed viral growth and mRNA translation and caused infected cells to be less likely to replicate. An abortive infection was caused by miR-28-3p expression, which inhibited HTLV-1 reverse transcription and prevented the formation of the preintegration complex [122].

The viral oncoprotein E6 of oncogenic HPV inhibited the expression of tumor-suppressing miR-34a [123]. Using miR-34a mimetics is expected to be effective in treating HPV-associated cancers.

## 5. Current Clinical Trials Targeting miRNAs in Oncogenic Viruses

Several clinical trials are presently underway that either employ miRNA directly or use it as an indicator of the response to the treatment of oncogenic viral infections. However, these studies are mostly directed toward host miRNAs. In a phase 2 clinical trial (NCT03923842), the effect of denosumab on EBV nasopharyngeal carcinoma will be assessed by evaluating the serum and salivary levels of nasopharyngeal carcinoma-associated miRNAs. Similarly, miR-122, a liver-specific miRNA, will be used to follow the response to the treatment of HCV infections (NCT00980161).

It was reported that individuals with hypertension had elevated levels of HCMV-miR-UL112, an miRNA encoded by HCMV. The results showed that IRF-1 was a direct target gene of HCMV-miR-UL112, which was predicted to act on many other genes. A current clinical trial (NCT01113359) is investigating the contribution of HCMV to hypertension.

## 6. Conclusions

Whilst the roles of v-miRNAs are only now becoming known, v-miRNAs cannot only target viral transcripts, but also cellular transcripts. V-miRNAs interact with other viral macromolecules to reprogram host cells in order to regulate the transition between the dormant and lytic stages of a virus, boost viral replication by boosting cell survival, and antagonize immunological responses. Modulation of the environment of the host cell is achieved through a combination of processes, some of which are redundant. This is made feasible by the combined efforts of v-miRNAs and proteins, which create a cellular environment that allows the virus to replicate and spread. Therefore, V-miRNAs provide remarkable therapeutic benefits and are considered to be intriguing antivirals that could be used as miRNA-based antivirals. V-miRNAs are the most promising technique currently available for assessing viral infectiousness and reproductive capacity because of their extensive diagnostic and prognostic value as biomarkers. The research on v-miRNAs is a rapidly growing science. When the mechanism underlying the activity of v-miRNAs has been better defined and elucidated, v-miRNAs may be utilized for the early detection and control of virally induced cancers.

## Figures and Tables

**Figure 1 pharmaceuticals-16-00485-f001:**
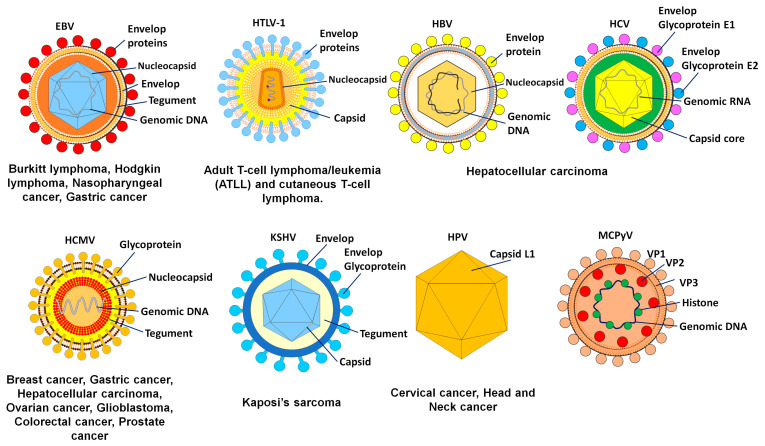
The oncogenic viruses with their target tissues and consequent malignancies.

**Figure 2 pharmaceuticals-16-00485-f002:**
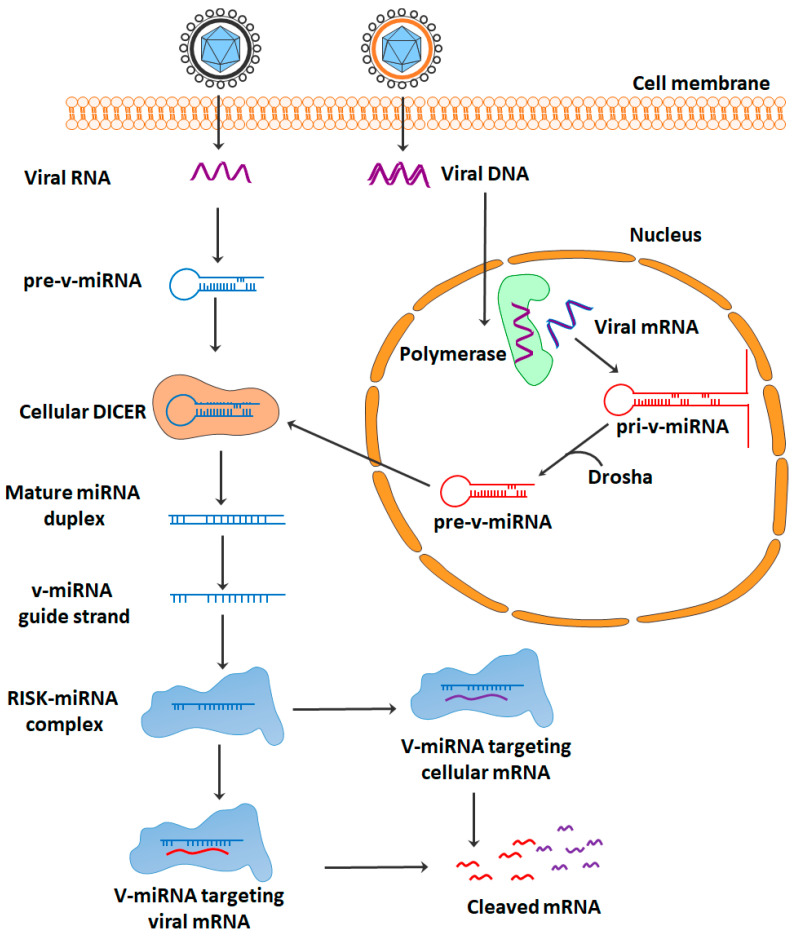
Schematic overview of the mechanism of action of v-miRNAs.

**Figure 3 pharmaceuticals-16-00485-f003:**
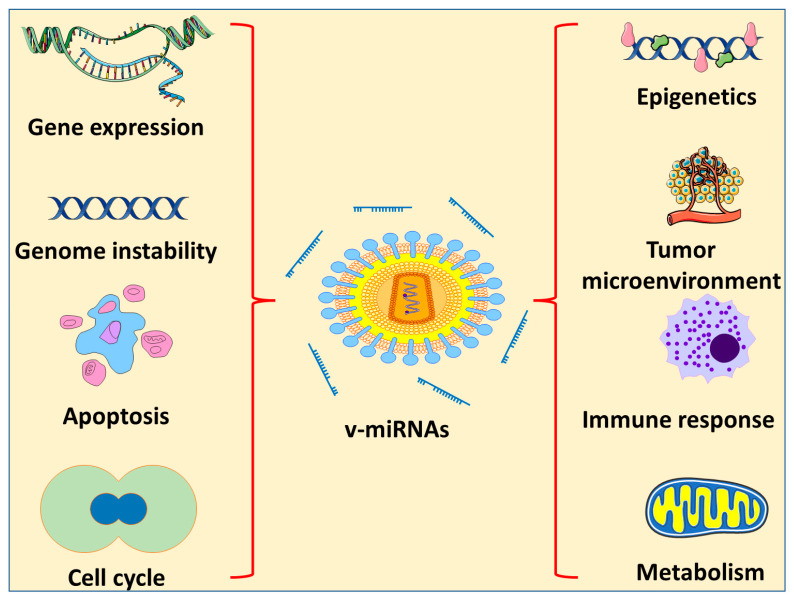
Effects of v-miRNA on different characteristics of cancer progression.

**Table 1 pharmaceuticals-16-00485-t001:** v-miRNAs, their targets, and the malignancies with which they are involved.

Viral miRNA	Virus	Target	Type of Cancer	Reference
miR-BART9	EBV	E-Cadherin	Nasopharyngeal carcinoma	[11]
miR-BART1	EBV	PTEN	Nasopharyngeal carcinoma	[12]
miR-BART10-3p	EBV	ALK7	Nasopharyngeal carcinoma	[13]
miR-BART20-5p	EBV	NDRG1	Gastric cancer	[14]
miR-BART2-5p and miR-BART11-5p	EBV	RB and p21	Gastric cancer	[15]
miR-BART5-5p	EBV	PIAS3	Gastric cancer	[16]
miR-BART11	EBV	FOXP1	Gastric cancer	[17]
miR-BART18-3p	EBV	Sirtuin	Colorectal cancer	[18]
miR-US4-1	CMV	ERAP1	HeLa human cervical cancer	[19]
CMV70-3P	CMV	SOX2	Glioma	[20]
miR-K12-3	KSHV	GRK2	Renal cancer	[21]
miR-K12-1-5p	KSHV	SOCS6	Kaposi sarcoma	[22]
miR-H1-1	HPV	Epithelium development (RGMA, SHANK3, PAX6, PFN1, WNT4), cell migration (CAV2, ITGAM, PAX6, PTEN, SEMA3F, ULK1), focal adhesion (CAV2, IGF1R, ITGB8, PTEN, PIK3CD), and cancer (CBL, CYCS, FGF7, IGF1R, PTEN, PIK3CD, WNT4).	HPV 16-associated cancers	[23]
miR-H2-1	HPV	PKNOX1, SP3, XRCC4	HPV-associated cancers	[23]
miR-3	HBV	PPM1A	Hepatocellular carcinoma	[24]
miR-2	HBV	TRIM35	Hepatocellular carcinoma	[25]

## Data Availability

Not applicable.

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
