# Peer review of "Oncogenic Viruses-Encoded microRNAs and Their Role in the Progression of Cancer: Emerging Targets for Antiviral and Anticancer Therapies"

_pharmaceuticals, 2023, doi:10.3390/ph16040485_

Round 1

Reviewer 1 Report

In the Manuscript (MS) by Kandeel the author made a revision of literature data on the role of viral derived microRNA (v-miRNAs) in progression cancer progression. Fist of all, author introduced the oncogenic features of human oncogenic viruses from Herpesviruses to KSHV, then he described for each viral family, the nature and the reported effects of v-miRNAs in the tumorigenesis process with a particular emphasis on the ability of v-miRNAs to deregulate proliferation and migratory capacity of infected cells as well as to influence epithelial-mesenchymal transition (EMT). Finally, the author report possible therapeutic approaches aimed to revert miRNA’s deregulation in cancer cells.

The MS is well conceived, and it was linearly developed, and I appreciate the idea of focus the topic of the review on v-miRNAs rather than on cellular ones (c-miRNAs) deregulated following viral infection. Nevertheless, some revisions need to be applied to make the review suitable for publication in “Pharmaceuticals”:

1. In paragraph 1, author listed the seven human oncogenic viruses but when he described their features in paragraph 2, MCPyV is missed. Same for Fig. 1 where HIV-1 is referred as “oncogenic”, whereas HTLV-1 is missed. I suggest the author to add a brief description of MCPyV tumorigenesis in paragraph 2 and to modify figure 1 by adding MCPyV picture and by changing HIV-1 into HTLV-1.

Regarding the v-miRNAs possibly codified by MCPyV, if they are unknown, the author could briefly report in the paragraph 2 the lack of published data, otherwise he must add an additional paragraph in the section 3 about their role in MCPyV tumorigenesis, as he did for all the other oncoviruses.

2. In paragraph 3.4 (HPV), author described the role of (cellular) miR-20 and -21 as potential therapeutic biomarkers (lines 373-376). As I’ve already stated the strength of the MS is to be centered on v-miRNAs, so I suggest the author to remove the relevant sentence.

3. In paragraph 3.5 (HBV, HCV), author report the role of two cellular miRNAs (i.e. miR-122 and miR-199a*) as modulators of HCV genomic expression (Lines 404-417). Even if these miRNAs are, again, off-topic being c-miRNAs, in this case I suggest the author to hold the relevant part of the paragraph by modifying the introductory sentence, underlying the importance, in this particular case, of c-miRNAs for HCV genomic expression.

Minor spelling revision are as follow:

Line 106: probably a ref annotation is uncorretly reported (i.e. … NPC have a positive EBV test,13 and a close connection between… should be …NPC have a positive EBV test [13], and a close connection between…). Check and revise if necessary.

Line 143: Same as above

Lines 123-24: “Gene products from human cytomegalovirus (HCMV) also target other cellular functions crucial to tumor formation, such as: Also….”: It looks to me that something was missing before the word “Also”. Check and revise if needed.

Lines 345-346: Check the spelling of “IB/NF-B signalling”: it looks to me something is possibly missed.

Lines 371-2: Check the sentence “It is interesting to note that this gene controls cell migration and proliferation and binds to and triggers the E6 and E7 promoters of early HPV 16”: which gene? Is it TEF-1 reported in ref 70?

Lines 425-26: In the sentence “In this context, degradation of the overexpressed mRNA by using miRNA…” is the author rather referring to anti-miRNA? Check and revise if needed.

Author Response

Reviewer 1

In the Manuscript (MS) by Kandeel the author made a revision of literature data on the role of viral derived microRNA (v-miRNAs) in progression cancer progression. Fist of all, author introduced the oncogenic features of human oncogenic viruses from Herpesviruses to KSHV, then he described for each viral family, the nature and the reported effects of v-miRNAs in the tumorigenesis process with a particular emphasis on the ability of v-miRNAs to deregulate proliferation and migratory capacity of infected cells as well as to influence epithelial-mesenchymal transition (EMT). Finally, the author report possible therapeutic approaches aimed to revert miRNA’s deregulation in cancer cells.

The MS is well conceived, and it was linearly developed, and I appreciate the idea of focus the topic of the review on v-miRNAs rather than on cellular ones (c-miRNAs) deregulated following viral infection. Nevertheless, some revisions need to be applied to make the review suitable for publication in “Pharmaceuticals”:

Response

The positive response is highly appreciated.

  1. In paragraph 1, author listed the seven human oncogenic viruses but when he described their features in paragraph 2, MCPyV is missed. Same for Fig. 1 where HIV-1 is referred as “oncogenic”, whereas HTLV-1 is missed. I suggest the author to add a brief description of MCPyV tumorigenesis in paragraph 2 and to modify figure 1 by adding MCPyV picture and by changing HIV-1 into HTLV-1.

Response

Thanks to the reviewer for this comment. Figure 1 is modified as instructed.

A description of the viruses is given in 2 paragraphs at section 2. The data related to their miRNA is provided in section 3.5.

While MCPyV and HTLV-1 are oncogenic viruses, it was reported that they are either not encoding viral miRNAs or one miRNA with low final outcomes.

Owing to the deficiency in this paper, the text was enriched with further details about these two viruses. The new data included the biological aspects as well as their effect on miRNAs.

The added text as follows:

1-at the end of section 2

MCV is thought to be the cause of Merkel cell carcinoma, which is an uncommon but aggressive form of skin cancer [54]. The sequencing of the virus recovered from Merkel cell tumours contains tumor-specific mutations that terminate the MCV T antigen. These mutations (which are not present in wild-type virus recovered from nontumor sites) inactivate the T antigen helicase, leaving the integrated virus incapable of replicating outside of the host cancer cell. As a result, the tumor acts as a "dead-end host" for MCV [55].

HTLV-1 is a retrovirus associated with several diseases, comprising lymphomas and myelopathy. It is predicated that about 1-5% of HTLV-1-infected cases develop cancers [56] e.g. adult T-cell lymphoma/leukemia (ATLL) and cutaneous T-cell lymphoma.

2-the newly added section 3.5

3.5. HTLV-1 and MCPyV

HTLV-1 oncogenic mechanisms comprised the direct impact of viral proteins on the host miRNA machinery. In this regard, HTLV-1-encoded miRNAs have not been reported [87,88]. HTLV-1 increases the expression of host cells miRNAs involved in proliferation and apoptosis [88]. Comparison of the miRNA expression profiles in normal and HTLV-1-infected hosts revealed several deregulated miRNA as miR-29c, miR-30c, miR-193a-5p and miR-885-5p, which can be regarded as diagnostic tools for HTLV-1 infection [89]. It was reported that about 22 upregulated and 22 downregulated miRNAs were identified in acute ATLL patients [87].

Reports have indicated that MCPyV miRNAs are either not present or present at extremely low levels in malignancies [90]. MCV-miR-M1-5p, which is MCPyV-encoded miRNA was found to be expressed at low levels in about a half of the virus positive cases. Interestingly, mcv-miR-M1 has been associated with negative regulation of the viral DNA expression [91].

Regarding the v-miRNAs possibly codified by MCPyV, if they are unknown, the author could briefly report in the paragraph 2 the lack of published data, otherwise he must add an additional paragraph in the section 3 about their role in MCPyV tumorigenesis, as he did for all the other oncoviruses.

Response

These sections were modified as instructed. Please review the track changes file.

  1. In paragraph 3.4 (HPV), author described the role of (cellular) miR-20 and -21 as potential therapeutic biomarkers (lines 373-376). As I’ve already stated the strength of the MS is to be centered on v-miRNAs, so I suggest the author to remove the relevant sentence.

Response

This comment is highly appreciated.

The miRNAs that nonrelevant to the virus-encoded miRNAs were removed.

  1. In paragraph 3.5 (HBV, HCV), author report the role of two cellular miRNAs (i.e. miR-122 and miR-199a*) as modulators of HCV genomic expression (Lines 404-417). Even if these miRNAs are, again, off-topic being c-miRNAs, in this case I suggest the author to hold the relevant part of the paragraph by modifying the introductory sentence, underlying the importance, in this particular case, of c-miRNAs for HCV genomic expression.

Response

This note also is highly appreciated.

The text of this section was modified as follows:

“In cases of HCV, the host miRNAs plays a central role in virus replication and constitute a valid antiviral target. Initially, the role for miR-122 in HCV infection was indicated by sequestering endogenous miR-122, which led to a significant decrease in HCV RNA in human liver cells that contain HCV replicons………………………”

Minor spelling revision are as follow:

Line 106: probably a ref annotation is uncorretly reported (i.e. … NPC have a positive EBV test,13 and a close connection between… should be …NPC have a positive EBV test [13], and a close connection between…). Check and revise if necessary.

Response

Corrected

Line 143: Same as above

Response

Corrected

Lines 123-24: “Gene products from human cytomegalovirus (HCMV) also target other cellular functions crucial to tumor formation, such as: Also….”: It looks to me that something was missing before the word “Also”. Check and revise if needed.

Response

This sentence is deleted. It is a repetition of the previous sentence.

Lines 345-346: Check the spelling of “IB/NF-B signalling”: it looks to me something is possibly missed.

Response

Thank you for this comment

It is corrected to IκBα/NF-κB

Lines 371-2: Check the sentence “It is interesting to note that this gene controls cell migration and proliferation and binds to and triggers the E6 and E7 promoters of early HPV 16”: which gene? Is it TEF-1 reported in ref 70?

Response

The text was modified as follows

“It is interesting to note that the HPV 16 transcriptional enhancer factor (TEF-1) controls cell migration and proliferation and binds to and triggers the E6 and E7 promoters of early HPV 16 [76]. In this context, TEF-1 and its associated factors are essential for HPV 16 transcription.”

Lines 425-26: In the sentence “In this context, degradation of the overexpressed mRNA by using miRNA…” is the author rather referring to anti-miRNA? Check and revise if needed.

Response

The sentence is corrected as follows

“In this context, MiRNA expression alterations can be dealt with in two primary ways: 1) degradation of the overexpressed mRNA via miRNA, and 2) regulation of miRNA expression. Injecting miRNAs to make up for miRNAs that aren't being produced enough of is the second option.”

Reviewer 2 Report

The present review reports about oncogenic viruses and their role in promoting human neoplasms and advancement of malignancies via the expression of viral miRNAs. The review also discusses potential therapies against this infection and malignancies. 

Following are the comments to highlight certain limitations after review of the paper;

1.     Line no 11 – Kindly replace stimulated with caused.

2.  Line no 15-16; 47-49 - Abstract points out the role of Merkel Cell polyomavirus (MCPyV) and HTLV-1. There is no description about the same in the review. Authors are requested to add the miRNAs associated with these virus groups.

3.     Line no 33-35 – Kindly change the sentence description according to the proper grammar.

4.     Figure 1 – Kindly elaborate the schematics associated with the virus structure with proper labelling. 

5.     Line no 62 – Kindly explain about the concept associated with using orthologues of viral and host miRNAs.

6.     Figure 2 – How does the direct maturation of pre-v-miRNA occurs in the cytoplasm? What is the process employed in intranuclear processing? Kindly update the figure correctly. Kindly explain about the same in the writing part.

7.     Line no 89 – Kindly replace nominated with “known”.

8.     Line no 106 – Remove 13 from the line.

9.   Line no 111 – The word diagnoses do not represent the data correctly, kindly replace with incidence.

10.  Line no 115 – Kindly elaborate a bit about biological role of EBNAs.

11.  Line no 124 – The sentence is incomplete after such as; kindly complete by adding the cellular function essential to tumor formation.

12.  Line no 139 – Kindly replace main with major.

13.  Line no 143 – Remove 5 from the line.

14.  Line no 144 – Mention the two kinds of HPV responsible for majority of HPV related malignancies.

15.  Table 1 – miR-US4-1 role in cervical cancer is shown. The reference however showed that the cell lines used for the study was HEK293T and HeLa, both. Is it justifiable to designate this miRNA for cervical cancer?

16.  Table 1 – The target for miR-H1-1 represents host system modulation and immunological modulations. Kindly add which targets are involved in neoplastic development and cellular transformation?

17.  Table 1 - miR-H1-1 & miR-H2-1 belong to which specific HPV subtype? The cancer associated with the same are primarily which type of HPV associated cancers?

18.  Line no 244-245 – Does the sentence “BART1’s reduction in PTEN dosage induced EMT” clearly represent the fact? The reference states PTEN dosage reduction by BART1. Kindly check. Which PTEN dependent pathways are activated by the same?

19.  Line no 257-261 – miR-BART5-5p controls the miR-BART5/PIAS3/pSTAT3/PD-L1 axis by enhancing PD-L1. The mechanism happens in PD-L1 positive tumors. What is the role of this miRNA in PD-L1 negative tumors?

20.  Line no 262-267 – How FOXP3 downregulation promote EMT in tumor microenvironment?

21.  The role of EBV associated viral miRNAs is not discussed with respect to major malignancies like Burkitt’s lymphoma and Hodgkin’s lymphoma? Any reason for the same.

22.  Line no 281 – Kindly correct the miRNA name – CMV70-3P.

23.  Section 3.2 HCMV – The references reviewed and cited showed HCMV associated miRNA role in latency but how this miRNA is involved in the progression of malignancies and HCMV associated cancers ( type of cancer) has not been discussed.

24.  Section 3.3 KSHV – Author have reviewed about the KSHV associated miRNA generated from 12 stem loop structures. In the next reference, are these the same miRNA involved in release of VEGF4 protein and upregulation of MMP involved in angiogenesis ? 

25.  Section 3.3 KSHV – Is there any difference in the miRNA regulation depending upon the stage of infection as KSHV associated miRNA are both expressed during latent and lytic cycle of the virus ?

26.  Line no 314 – The repression of GADD45B is also mediated by hypermethylation of the promoter. Does it have any significance on the KSHV associated malignancies? Is it possible that the suppression of GADD45B in the current scenario is mediated by hypermethylation rather than viral miRNA?

27.  Line no 336 – Is there any specific KSHV associated cancer where mTORC1 pathway is deregulated and targeted? 

28.  Section 3.4 HPV – Authors have referenced the role of HPV-41- miRNA and FcPV1-miRNA in HPV life cycle. HPV-41 is not known to be a part of high-risk HPV and HPV associated malignancies. Is it a proper reference to be incorporated under the section of viral miRNA and cancers?

29.  Section 3.5 Hepatitis viruses (HBV, HCV) – Line no 384 – Reference reviewed (Yao et.al) has been retracted. Authors are requested to add other reference in the section.

30.  Section 3.5 Hepatitis viruses (HBV, HCV) – The section does not represent viral miRNA associated with HBV and HCV associated malignancies, rather the host miRNAs are primarily discussed. The section needs major revision and new references need to be incorporated which identify the role of viral miRNA in HBV & HCV associated malignancies.

31.  Line no 428 – Kindly write the referencing correctly as Yang et.al. 

32.  Section Anti-MiR & MiR mimetics – Limited examples and literature review has been presented in these sections which are basically associated with HCV. Since the review paper encompasses major oncogenic viruses, the referencing and literature review is needed to be provided for all.

33.  Line no 458 – Correct miRMA to miRNA.

34.  Section 5 – Current Clinical Trials – The clinical trials associated with miRNA involved in viral infection mediated cancers need to be incorporated.  

The review article presented gives an idea about multiple aspects associated with oncogenic viruses and their role in malignancies. However, the data presented need to be updated as per the points mentioned. The data represented in therapy section need to be reviewed extensively and new references need to be studied and incorporated to justify the large scope of paper. Multiple studies and data have been generated with respect to oncogenic viruses and the associated viral miRNA. Authors need to review the literature more to fulfill the gaps identified in the current manuscript.

Author Response

Reviewer 2

The present review reports about oncogenic viruses and their role in promoting human neoplasms and advancement of malignancies via the expression of viral miRNAs. The review also discusses potential therapies against this infection and malignancies. 

Following are the comments to highlight certain limitations after reviewing of the paper;

  1. Line no 11 – Kindly replace stimulated with caused.

Response

Replaced

  1. Line no 15-16; 47-49 - Abstract points out the role of Merkel Cell polyomavirus (MCPyV) and HTLV-1. There is no description about the same in the review. Authors are requested to add the miRNAs associated with these virus groups.

Response

A description of the viruses is given in 2 paragraphs in section 2. The data relating to their miRNA is provided in section 3.5.

Thanks to the reviewer for this comment. While MCPyV and HTLV-1 are oncogenic viruses, it was reported that they are either not encoding viral miRNAs or one miRNA with low final outcomes.

Owing to the deficiency in this paper, the text was enriched with further details about these two viruses. The new data included the biological aspects as well as their effect on miRNAs.

The added text as follows:

1-at the end of section 2

MCV is thought to be the cause of Merkel cell carcinoma, which is an uncommon but aggressive form of skin cancer [1]. The sequencing of the virus recovered from Merkel cell tumours contains tumor-specific mutations that terminate the MCV T antigen. These mutations (which are not present in wild-type viruses recovered from nontumor sites) inactivate the T antigen helicase, leaving the integrated virus incapable of replicating outside of the host cancer cell. As a result, the tumor acts as a "dead-end host" for MCV [2].

HTLV-1 is a retrovirus associated with several diseases, comprising lymphomas and myelopathy. It is predicted that about 1-5% of HTLV-1-infected cases develop cancers [3] e.g. adult T-cell lymphoma/leukemia (ATLL) and cutaneous T-cell lymphoma.

2-the newly added section 3.5

3.5. HTLV-1 and MCPyV

HTLV-1 oncogenic mechanisms comprised the direct impact of viral proteins on the host miRNA machinery. In this regard, HTLV-1-encoded miRNAs have not been reported [4,5]. HTLV-1 increases the expression of host cells miRNAs involved in proliferation and apoptosis [5]. Comparison of the miRNA expression profiles in normal and HTLV-1-infected hosts revealed several deregulated miRNA as miR-29c, miR-30c, miR-193a-5p and miR-885-5p, which can be regarded as diagnostic tools for HTLV-1 infection [6]. It was reported that about 22 upregulated and 22 downregulated miRNAs were identified in acute ATLL patients [4].

Reports have indicated that MCPyV miRNAs are either not present or present at extremely low levels in malignancies [7]. MCV-miR-M1-5p, which is MCPyV-encoded miRNA was found to be expressed at low levels in about half of the virus-positive cases. Interestingly, mcv-miR-M1 has been associated with negative regulation of viral DNA expression [8].

  1. Line no 33-35 – Kindly change the sentence description according to the proper grammar.

Response

Corrected and the paragraph is modified.

After editing the manuscript was sent to an English language editing service for checking by a native English speaker.

  1. Figure 1 – Kindly elaborate the schematics associated with the virus structure with proper labelling. 

Response

A new figure was developed with updated labeling and clarification.

  1. Line no 62 – Kindly explain about the concept associated with using orthologues of viral and host miRNAs.

Response

The information is updated and a description of orthologues is given.

  1. Figure 2 – How does the direct maturation of pre-v-miRNA occurs in the cytoplasm? What is the process employed in intranuclear processing? Kindly update the figure correctly. Kindly explain about the same in the writing part.

Response

A new figure was produced and an explanation of the content is given in the text.

  1. Line no 89 – Kindly replace nominated with “known”.

Response

Corrected

  1. Line no 106 – Remove 13 from the line.

Response

Corrected

  1. Line no 111 – The word diagnoses do not represent the data correctly, kindly replace with incidence.

Response

Corrected

  1. Line no 115 – Kindly elaborate a bit about biological role of EBNAs.

Response

The biological aspects were added. The added text s as below

“Epstein-Barr nuclear antigens (EBNAs) 1, 2, 3A, 3B, 3C, and LP are expressed during latent infection in host cells [9]. EBNAs are multifunctional proteins and are considered the only proteins expressed in all types of EBV and found in EBV-associated malignancies [10]. The positive and negative regulation of viral promoters by EBNA1 is essential for gene control, extrachromosomal replication, and maintenance of the EBV episomal genome [11].”

  1. Line no 124 – The sentence is incomplete after such as; kindly complete by adding the cellular function essential to tumor formation.

Response

Thank you for this comment

This sentence is deleted. It is a repetition of the previous sentence.

  1. Line no 139 – Kindly replace main with major.

Response

Corrected

  1. Line no 143 – Remove 5 from the line.

Response

Corrected

  1. Line no 144 – Mention the two kinds of HPV responsible for majority of HPV related malignancies.

Response

The sentence is modified as follows

“There are two kinds of HPV that are responsible for the majority of HPV-related malignancies (productive (CIN1) or abortive (CIN3)), including about 70% of cervical cancers and pre-cancerous cervical lesion”

  1. Table 1 – miR-US4-1 role in cervical cancer is shown. The reference however showed that the cell lines used for the study was HEK293T and HeLa, both. Is it justifiable to designate this miRNA for cervical cancer?

Response

Thank you so much for this important comment.

The text has been updated to indicate “HeLa human cervical cancer cells”, which is reported by the authors.

  1. Table 1 – The target for miR-H1-1 represents host system modulation and immunological modulations. Kindly add which targets are involved in neoplastic development and cellular transformation?

Response

The targets are added as follows

“Epithelium development (RGMA, SHANK3, PAX6, PFN1, WNT4), cell migration (CAV2, ITGAM, PAX6, PTEN, SEMA3F, ULK1), focal adhesion (CAV2, IGF1R, ITGB8, PTEN, PIK3CD), and cancer (CBL, CYCS, FGF7, IGF1R, PTEN, PIK3CD, WNT4).”

  1. Table 1 - miR-H1-1 & miR-H2-1 belong to which specific HPV subtype? Cancer associated with the same are primarily which type of HPV-associated cancers?

Response

HPV 16 is updated in the text

  1. Line no 244-245 – Does the sentence “BART1’s reduction in PTEN dosage induced EMT” clearly represent the fact? The reference states PTEN dosage reduction by BART1. Kindly check. Which PTEN dependent pathways are activated by the same?

Response

The text has been modified and corrected

“Reduction in PTEN dosage by BART1 activated PTEN-dependent pathways, induced EMT and boosted nasopharyngeal cancer migration, invasion, and metastasis”

  1. Line no 257-261 – miR-BART5-5p controls the miR-BART5/PIAS3/pSTAT3/PD-L1 axis by enhancing PD-L1. The mechanism happens in PD-L1-positive tumors. What is the role of this miRNA in PD-L1 negative tumors?

Response

PIAS3 reduction was directly driven by miR-BART5-5p.

overall survival in EBV-associated gastric carcinomas was significantly shortened for those with PD-L1(+) tumors compared to those with PD-L1(−). This indicates the general efficiency of miR-BART5-5p in PD-L1(+) tumors, and low efficiency in PD-L1(−), implying the importance of PD-L1(+) in the overall actions of miR-BART5-5p. based on this, miR-BART5-5p is suggested to be PD-1/PD-L1 immune checkpoint inhibitor therapy.

The text is modified to indicate this.

  1. Line no 262-267 – How FOXP3 downregulation promote EMT in tumor microenvironment?

Response

The text is modified to include the potential mechanism of FOXP3-induced EMT as follows.

“In addition, the EBV-encoded viral miRNA-BART11 downregulates the FOXP1 transcription factor and promotes EMT through acting on gastric tumor cells and the tumor microenvironment. This could hasten the invasion and spread of cancer, which would have repercussions for patient survival and prognosis [12]. Reduced levels of FOXP1 led to changes in the expression of the EMT transcription factors E-cadherin and snail. The EBV-miR-BART11-FOXP1 signaling axis can enhance the activity of EMT inducers NF-κB and IL-6 through the downregulation of E-cadherin.”

  1. The role of EBV associated viral miRNAs is not discussed with respect to major malignancies like Burkitt’s lymphoma and Hodgkin’s lymphoma? Any reason for the same.

Response

The role of miRNAs on Burkitt’s lymphoma and Hodgkin’s lymphoma is now updated. The following two paragraphs were added.

“There was an increase in viral-miRNAs in BL patients. The v-miRNAs BART1–3p, BART3, BART4, BART6–3p, BART6–5p, BART7, BART8, BART9, BART10 and 12 other EBV miRNAs were upregulated in BL. These results bolstered the theory that the expression pattern of EBV-related miRNAs in BL is disease-specific and offered insight into the possible function of EBV-encoded miRNA in BL [13]. miR-BART3 was found to target the importin 7 receptor, which shares in transportation of transctiotion factors to the nucleus [14]. The miR-BART6-3p can target the host immune response by ibterfereng with type I IFN pathways [15]. The miR-BART2-5p prevents the virus from being attacked by immune cells by inhibiting the MHC-class 1 chain-related molecule B [16].

Several miRNAs were found to be associated with the progression of HL. miR-BART2-5p is upregulated in HL, inhibited NF-kB activation and help in immune escape of cancer cells [17]. The exosomal release of miR-BART13-3p, the most common miR-BART in Hodgkin lymphoma, has been shown to have a role in the development of the disease [18].”

  1. Line no 281 – Kindly correct the miRNA name – CMV70-3P.

Response

Corrected

  1. Section 3.2 HCMV – The references reviewed and cited showed HCMV associated miRNA role in latency but how this miRNA is involved in the progression of malignancies and HCMV associated cancers ( type of cancer) has not been discussed.

Response

The text was modified to include more details about this part

“……..Due to their ability to silence human genes involved in a variety of physiological processes, HCMV-miRNAs have lately emerged as a prospective clinical diagnosis for a wide range of human disorders [19]. Initial reports showed that HCMV is not a virus that causes cancer, it seems likely that HCMV will profit from the degradation of these host miRNAs for reasons unrelated to oncogenesis [20]. However, oncogenic characteristics are possessed by multiple members of the HCMV proteome [21]. It has been demonstrated that the expression of of HCMV chemokine receptor US28 in NIH3T3 cells results in transformed phenotypes and tumor formation [22]. The HCMV UL76 protein has been also associated with chromosomal breaks in human glioblastoma cells [23]……….”

  1. Section 3.3 KSHV – Author have reviewed about the KSHV associated miRNA generated from 12 stem loop structures. In the next reference, are these the same miRNA involved in release of VEGF4 protein and upregulation of MMP involved in angiogenesis ? 

Response

Both studies used the miRNA sets that are already expressed from KSHV. The actual number of KSHV miRNAs exceeds that of the 12 stem-loop precursors (about 25), due to the expression of relatively abundant 5p and 3p miRNAs from several of the pre-miRNAs. However, the expression levels were too into account in the first study.

  1. Section 3.3 KSHV – Is there any difference in the miRNA regulation depending upon the stage of infection as KSHV associated miRNA are both expressed during latent and lytic cycle of the virus ?

Response

More details are added for explanation of miRNA regulation in KSHV as follows:

“”The human pathogenic γ-herpesvirus known as KSHV is closely linked to the occurrence of primary effusion lymphoma as well as KS and B cell proliferative diseases. A topic of rapidly growing importance is the documentation and mechanistical analysis of non-coding RNAs expressed by KSHV. miRNAs are expressed by KSHV during both the latent and lytic phases of infection [24,25]. The control of these miRNAs may change at various points in the course of an infection. KSHV expresses a subset of miRNAs during the latent phase of infection that aid the virus in evading the host immune response and staying dormant. The viral latency-associated nuclear antigen (LANA) protein regulates these miRNAs by binding to the promoter regions of the miRNA genes [26]. A distinct set of miRNAs implicated in viral replication and pathogenicity is expressed by KSHV during the lytic phase of infection [25]. The viral immediate-early gene product lytic replication and transcription activator (RTA) controls the expression of miRNAs by binding to promoter sites and turning them on. Therefore, pathways that either promote or inhibit lytic replication are crucial to the pathogenicity of KSHV. The RTA, together with LANA, are thought to control this delicate equilibrium [27].

The miRNAs associated with KSHV may be regulated in a manner that varies with the severity of the infection. To completely comprehend the mechanisms driving miRNA regulation during KSHV infection more research is required.

  1. Line no 314 – The repression of GADD45B is also mediated by hypermethylation of the promoter. Does it have any significance on the KSHV associated malignancies? Is it possible that the suppression of GADD45B in the current scenario is mediated by hypermethylation rather than viral miRNA?

Response

It is probable that KSHV-associated malignancies are influenced by the hypermethylation-mediated inhibition of GADD45B. GADD45B silencing can be achieved through hypermethylation of its promoter. GADD45B is a tumour suppressor gene involved in cell cycle control, DNA repair, and programmed cell death. Repression of this gene has been linked to the development of KSHV. GADD45B is repressed in KSHV-associated tumours.

The inhibition of GADD45B in KSHV-associated cancers may be mediated by hypermethylation of the promoter, even though viral miRNAs have been demonstrated to influence the expression of numerous genes in similar diseases. Further study is required to determine the relative contributions of viral miRNAs and hypermethylation in the regulation of GADD45B in KSHV-associated malignancies.

  1. Line no 336 – Is there any specific KSHV associated cancer where mTORC1 pathway is deregulated and targeted? 

Response

A number of various cancers, such as breast, prostate, lung, and kidney cancer, have been found to have abnormal mTORC1 signaling. mTORC1 is a valid target for KS. The text is modified to indicate this as follows:

“A number of various cancers, such as breast, prostate, lung, and kidney cancer, have been found to have abnormal signaling of the mammalian target of rapamycin complex 1 (mTOR) [28]. mTORC1 is inhibited by the cytosolic arginine sensor for mTORC1 (CASTOR1 and CASTOR2). The oncogenic mTORC1 pathway is typically deregulated in KS. This pathway controls cell proliferation, survival, and metabolism…..”

  1. Section 3.4 HPV – Authors have referenced the role of HPV-41- miRNA and FcPV1-miRNA in HPV life cycle. HPV-41 is not known to be a part of high-risk HPV and HPV associated malignancies. Is it a proper reference to be incorporated under the section of viral miRNA and cancers?

Response

This comment is highly appreciated.

HPV-41 is removed.

  1. Section 3.5 Hepatitis viruses (HBV, HCV) – Line no 384 – Reference reviewed (Yao et.al) has been retracted. Authors are requested to add other reference in the section.

Response

This section has been removed. The cited work depended on the initial work by Yao et al.

  1. Section 3.5 Hepatitis viruses (HBV, HCV) – The section does not represent viral miRNA associated with HBV and HCV associated malignancies, rather the host miRNAs are primarily discussed. The section needs major revision and new references need to be incorporated which identify the role of viral miRNA in HBV & HCV associated malignancies.

Response

The text is modified to indicate the virus-encoded miRNAs

“A study by Yang et al. demonstrated that HBV-encoded miRNA3 (HBV-miR-3) is expressed in HBV-infected tissues. To control HBV replication, HBV-miR-3 is transcribed during infection. This miR-3 regulates HBV replication by inhibiting HBsAg and HBeAg transcript synthesis [29] as well as targeting and DIX domain containing 1 (DIXDC1) and protein phosphatase 1A (PPM1A) [30].

It is belived that HCV is not encoding v-miRNAs [31]. In HCV, the host miRNAs plays a central role in virus replication and constitute a valid antiviral target. Initially, the role for miR-122 in HCV infection was indicated……………”

  1. Line no 428 – Kindly write the referencing correctly as Yang et.al. 

Response

Corrected

  1. Section Anti-MiR & MiR mimetics – Limited examples and literature review has been presented in these sections which are basically associated with HCV. Since the review paper encompasses major oncogenic viruses, the referencing and literature review is needed to be provided for all.

Response

It is important to note that anti-miRs are still a relatively new technology, and further research is needed to fully understand their safety and efficacy in treating oncogenic viruses.

Many of the possible treatments based on miRNA technology are targeting the host miRNAs, especially in the field of miR mimetics, which mostly affects the expression patterns in viruses hosts. However, some few studies are now available for the virus specific miRNAs. These studies are scarce but now included in their relevant sections.

This section is now eedited and improved, please review the track changes file for details.

  1. Line no 458 – Correct miRMA to miRNA.

Response

Corrected

  1. Section 5 – Current Clinical Trials – The clinical trials associated with miRNA involved in viral infection mediated cancers need to be incorporated.  

Response

The site for registered clinical trials was browsed with several search terms. However, to the best of our knowledge, there are no current clinical trial for employing oncogenic viruses encoded miRNAs. This act is now being updated in text.

The review article presented gives an idea about multiple aspects associated with oncogenic viruses and their role in malignancies. However, the data presented need to be updated as per the points mentioned. The data represented in therapy section need to be reviewed extensively and new references need to be studied and incorporated to justify the large scope of paper. Multiple studies and data have been generated with respect to oncogenic viruses and the associated viral miRNA. Authors need to review the literature more to fulfill the gaps identified in the current manuscript.

Response

The review has now been edited and updated in several aspects as instructed by the reviewer, which really enriched the content. We hope now the new changes meet the reviewer's expectations.

  1. Rotondo, J.C.; Bononi, I.; Puozzo, A.; Govoni, M.; Foschi, V.; Lanza, G.; Gafà, R.; Gaboriaud, P.; Touzé, F.A.; Selvatici, R.; et al. Merkel Cell Carcinomas Arising in Autoimmune Disease Affected Patients Treated with Biologic Drugs, Including Anti-TNF. Clinical cancer research : an official journal of the American Association for Cancer Research 2017, 23, 3929-3934, doi:10.1158/1078-0432.ccr-16-2899.
  2. Becker, J.C.; Houben, R.; Ugurel, S.; Trefzer, U.; Pföhler, C.; Schrama, D. MC polyomavirus is frequently present in Merkel cell carcinoma of European patients. The Journal of investigative dermatology 2009, 129, 248-250, doi:10.1038/jid.2008.198.
  3. Verdonck, K.; González, E.; Van Dooren, S.; Vandamme, A.M.; Vanham, G.; Gotuzzo, E. Human T-lymphotropic virus 1: recent knowledge about an ancient infection. The Lancet. Infectious diseases 2007, 7, 266-281, doi:10.1016/s1473-3099(07)70081-6.
  4. Moles, R.; Nicot, C. The emerging role of miRNAs in HTLV-1 infection and ATLL pathogenesis. Viruses 2015, 7, 4047-4074.
  5. Bouzar, A.B.; Willems, L. How HTLV-1 may subvert miRNAs for persistence and transformation. Retrovirology 2008, 5, 101, doi:10.1186/1742-4690-5-101.
  6. Fayyad-Kazan, M.; ElDirani, R.; Hamade, E.; El Majzoub, R.; Akl, H.; Bitar, N.; Fayyad-Kazan, H.; Badran, B. Circulating miR-29c, miR-30c, miR-193a-5p and miR-885-5p: Novel potential biomarkers for HTLV-1 infection diagnosis. Infection, Genetics and Evolution 2019, 74, 103938.
  7. Lee, S.; Paulson, K.G.; Murchison, E.P.; Afanasiev, O.K.; Alkan, C.; Leonard, J.H.; Byrd, D.R.; Hannon, G.J.; Nghiem, P. Identification and validation of a novel mature microRNA encoded by the Merkel cell polyomavirus in human Merkel cell carcinomas. Journal of clinical virology : the official publication of the Pan American Society for Clinical Virology 2011, 52, 272-275, doi:10.1016/j.jcv.2011.08.012.
  8. Theiss, J.M.; Günther, T.; Alawi, M.; Neumann, F.; Tessmer, U.; Fischer, N.; Grundhoff, A. A Comprehensive Analysis of Replicating Merkel Cell Polyomavirus Genomes Delineates the Viral Transcription Program and Suggests a Role for mcv-miR-M1 in Episomal Persistence. PLoS Pathog 2015, 11, e1004974, doi:10.1371/journal.ppat.1004974.
  9. AlQarni, S.; Al-Sheikh, Y.; Campbell, D.; Drotar, M.; Hannigan, A.; Boyle, S.; Herzyk, P.; Kossenkov, A.; Armfield, K.; Jamieson, L. Lymphomas driven by Epstein–Barr virus nuclear antigen-1 (EBNA1) are dependant upon Mdm2. Oncogene 2018, 37, 3998-4012.
  10. Humme, S.; Reisbach, G.; Feederle, R.; Delecluse, H.J.; Bousset, K.; Hammerschmidt, W.; Schepers, A. The EBV nuclear antigen 1 (EBNA1) enhances B cell immortalization several thousandfold. Proceedings of the National Academy of Sciences of the United States of America 2003, 100, 10989-10994, doi:10.1073/pnas.1832776100.
  11. Duellman, S.J.; Thompson, K.L.; Coon, J.J.; Burgess, R.R. Phosphorylation sites of Epstein-Barr virus EBNA1 regulate its function. The Journal of general virology 2009, 90, 2251-2259, doi:10.1099/vir.0.012260-0.
  12. Song, Y.; Li, Q.; Liao, S.; Zhong, K.; Jin, Y.; Zeng, T. Epstein-Barr virus-encoded miR-BART11 promotes tumor-associated macrophage-induced epithelial-mesenchymal transition via targeting FOXP1 in gastric cancer. Virology 2020, 548, 6-16.
  13. Piccaluga, P.P.; Navari, M.; De Falco, G.; Ambrosio, M.R.; Lazzi, S.; Fuligni, F.; Bellan, C.; Rossi, M.; Sapienza, M.R.; Laginestra, M.A.; et al. Virus-encoded microRNA contributes to the molecular profile of EBV-positive Burkitt lymphomas. Oncotarget 2016, 7, 224-240, doi:10.18632/oncotarget.4399.
  14. Dölken, L.; Malterer, G.; Erhard, F.; Kothe, S.; Friedel, C.C.; Suffert, G.; Marcinowski, L.; Motsch, N.; Barth, S.; Beitzinger, M.; et al. Systematic analysis of viral and cellular microRNA targets in cells latently infected with human gamma-herpesviruses by RISC immunoprecipitation assay. Cell Host Microbe 2010, 7, 324-334, doi:10.1016/j.chom.2010.03.008.
  15. Lu, Y.; Qin, Z.; Wang, J.; Zheng, X.; Lu, J.; Zhang, X.; Wei, L.; Peng, Q.; Zheng, Y.; Ou, C.; et al. Epstein-Barr Virus miR-BART6-3p Inhibits the RIG-I Pathway. Journal of innate immunity 2017, 9, 574-586, doi:10.1159/000479749.
  16. Nachmani, D.; Stern-Ginossar, N.; Sarid, R.; Mandelboim, O. Diverse herpesvirus microRNAs target the stress-induced immune ligand MICB to escape recognition by natural killer cells. Cell Host Microbe 2009, 5, 376-385, doi:10.1016/j.chom.2009.03.003.
  17. van Eijndhoven, M.A.; Zijlstra, J.M.; Groenewegen, N.J.; Drees, E.E.; van Niele, S.; Baglio, S.R.; Koppers-Lalic, D.; van der Voorn, H.; Libregts, S.F.; Wauben, M.H.; et al. Plasma vesicle miRNAs for therapy response monitoring in Hodgkin lymphoma patients. JCI insight 2016, 1, e89631, doi:10.1172/jci.insight.89631.
  18. Ramayanti, O.; Verkuijlen, S.; Novianti, P.; Scheepbouwer, C.; Misovic, B.; Koppers-Lalic, D.; van Weering, J.; Beckers, L.; Adham, M.; Martorelli, D.; et al. Vesicle-bound EBV-BART13-3p miRNA in circulation distinguishes nasopharyngeal from other head and neck cancer and asymptomatic EBV-infections. International journal of cancer 2019, 144, 2555-2566, doi:10.1002/ijc.31967.
  19. Diggins, N.L.; Hancock, M.H. HCMV miRNA targets reveal important cellular pathways for viral replication, latency, and reactivation. Non-coding RNA 2018, 4, 29.
  20. Guo, Y.E.; Steitz, J.A. Virus meets host microRNA: the destroyer, the booster, the hijacker. Molecular and cellular biology 2014, 34, 3780-3787.
  21. Herbein, G.; Kumar, A. The oncogenic potential of human cytomegalovirus and breast cancer. Frontiers in oncology 2014, 4, 230.
  22. Maussang, D.; Verzijl, D.; van Walsum, M.; Leurs, R.; Holl, J.; Pleskoff, O.; Michel, D.; van Dongen, G.A.; Smit, M.J. Human cytomegalovirus-encoded chemokine receptor US28 promotes tumorigenesis. Proceedings of the National Academy of Sciences of the United States of America 2006, 103, 13068-13073, doi:10.1073/pnas.0604433103.
  23. Siew, V.-K.; Duh, C.-Y.; Wang, S.-K. Human cytomegalovirus UL76 induces chromosome aberrations. Journal of Biomedical Science 2009, 16, 107, doi:10.1186/1423-0127-16-107.
  24. Broussard, G.; Damania, B. Regulation of KSHV latency and lytic reactivation. Viruses 2020, 12, 1034.
  25. Kincaid, R.P.; Sullivan, C.S. Virus-encoded microRNAs: an overview and a look to the future. PLoS pathogens 2012, 8, e1003018.
  26. Uppal, T.; Banerjee, S.; Sun, Z.; Verma, S.C.; Robertson, E.S. KSHV LANA—the master regulator of KSHV latency. Viruses 2014, 6, 4961-4998.
  27. Lu, C.C.; Li, Z.; Chu, C.Y.; Feng, J.; Feng, J.; Sun, R.; Rana, T.M. MicroRNAs encoded by Kaposi's sarcoma‐associated herpesvirus regulate viral life cycle. EMBO reports 2010, 11, 784-790.
  28. Strimpakos, A.S.; Karapanagiotou, E.M.; Saif, M.W.; Syrigos, K.N. The role of mTOR in the management of solid tumors: an overview. Cancer treatment reviews 2009, 35, 148-159.
  29. Yang, X.; Li, H.; Sun, H.; Fan, H.; Hu, Y.; Liu, M.; Li, X.; Tang, H. Hepatitis B virus-encoded microRNA controls viral replication. Journal of virology 2017, 91, e01919-01916.
  30. Chavalit, T.; Nimsamer, P.; Sirivassanametha, K.; Anuntakarun, S.; Saengchoowong, S.; Tangkijvanich, P.; Payungporn, S. Hepatitis B virus-encoded microRNA (HBV-miR-3) regulates host gene PPM1A related to hepatocellular carcinoma. Microrna 2020, 9, 232-239.
  31. Li, H.; Jiang, J.D.; Peng, Z.G. MicroRNA-mediated interactions between host and hepatitis C virus. World J Gastroenterol 2016, 22, 1487-1496, doi:10.3748/wjg.v22.i4.1487.

Round 2

Reviewer 2 Report

Thank you for addressing all the concerns raised by this reviewer. The revised manuscript is well updated with all these addressals. As the author mentioned that the manuscript has been edited through an English language editing service through a native English speaker, this should be mentioned in the Acknowledgement.

Author Response

Thank you for addressing all the concerns raised by this reviewer. The revised manuscript is well updated with all these addressals. As the author mentioned that the manuscript has been edited through an English language editing service through a native English speaker, this should be mentioned in the Acknowledgement.

Response

The requested data is added to the acknowledgement.

The service is provided by servicescape.com

The added text is

“The author would like to thank Marjorie Toensing for checking the English language, proofreading and editing the manuscript.”